# Rapid Detection of *Acinetobacter baumannii* Suspension and Biofilm Nanomotion and Antibiotic Resistance Estimation

**DOI:** 10.3390/biomedicines12092034

**Published:** 2024-09-06

**Authors:** Svetlana N. Pleskova, Nikolay A. Bezrukov, Ekaterina D. Nikolaeva, Alexey V. Boryakov, Olga V. Kuzina

**Affiliations:** 1Research Laboratory of Scanning Probe Microscopy, Lobachevsky State University of Nizhny Novgorod, 603022 Nizhny Novgorod, Russia; nick_bezrukov@mail.ru (N.A.B.);; 2Department “Nanotechnology and Biotechnology”, Nizhny Novgorod State Technical University named after Alekseev R.E., 603155 Nizhny Novgorod, Russia

**Keywords:** rapid test, atomic force microscopy, bacterial metabolism, biofilm, oscillation, disk diffusion test, scanning electron microscopy, bacterial morphology

## Abstract

Objectives: To develop a system for the rapid detection of *Acinetobacter baumannii* 173-p1 antibiotic resistance (to ensure reliable fixation of bacteria on a cantilever without losing their nanomotion, to show that nanomotion is due to bacterial metabolism, to compare the nanomotion of bacteria in suspension form and inside of the biofilms), to study the sensitivity/resistance of *A. baumannii* 173-p1 to antibiotics (lincomycin, ceftriaxone and doxycycline) using the oscillation method of atomic force microscopy and to evaluate the sensitivity and speed of the method in comparison with the classical disk diffusion method. Methods: The oscillation mode of atomic force microscopy, scanning electron microscopy and the classical disk diffusion method were used for a complex parallel study of *A. baumannii* 173-p1 antibiotic resistance, which included testing of fixing agents (poly-L-lysine, rosin and fibronectin), comparison of bacterial metabolism in a set of media (normal saline solution, meat-peptone broth and lysogeny broth) and assessment of antibiotic sensitivity/resistance per se. Results: A method for express testing of *Acinetobacter baumannii* antibiotic resistance using AFM was developed; it is shown that bacterial nanomotion directly correlates with bacteria metabolic activity and that bacterial nanomotion is more easily detected in suspension form, rather than in biofilms. Conclusion: The express testing method gave results that are completely comparable with the classical disk diffusion test and with the results of morphology studies by the SEM method, but it significantly exceeded them in speed, allowing a conclusion to be made on the sensitivity/resistance of bacteria less than an hour after the start of the diagnostics.

## 1. Introduction

*Acinetobacter baumannii* has gained great clinical importance over the past 10 years, primarily due to its ability to rapidly acquire resistance genes to many antibiotics. It probably became possible due to 23 types of plasmids with a capacity from 2.7 to 94.4 kB, which were found in acinetobacteria [1]. Nowadays, *A. baumannii* comes to the forefront in the occurrence of severe complications in intensive care unit patients and a number of infectious syndromes in military personnel in armed conflict zones [2]. The rapid global spread of *A. baumannii* strains resistant to all β-lactams, including carbapenems, illustrates the ability of the microorganism to respond quickly to changes under selective pressure [3]. Currently, *A. baumannii* is one of the six most dangerous bacteria (also known as the ESKAPE pathogens: *Enterococcus faecium*, *Staphylococcus aureus*, *Klebsiella pneumoniae*, *Acinetobacter baumannii*, *Pseudomonas aeruginosa* and *Enterobacter species*) causing nosocomial infections in the populations of developed countries [4]. For example, in Russian hospitals in 2012 the proportion of *Acinetobacter* spp. among all pathogens that caused postoperative purulent-inflammatory complications was 3.4% [5]. An analysis of complications of modern combat trauma among American soldiers in Iraq showed that *Acinetobacter* occupied first place among pathogens of wound infections, standing out in 36% of cases [6]. Such statistics of *Acinetobacter*-isolated strains’ high virulence turned out to be so unexpected that military doctors gave them the name “Iraqibacter” [7]. The high clinical significance of the pathogen necessitates the creation of systems for rapid detection of *A. baumannii* antibiotic resistance. The ability of *Acinetobacter* strains to form biofilms in human organism for sustainable survival (even with antibiotic therapy) is of particular importance [8]. Pili are the main adhesins involved in the anchoring of *Acinetobacter* cells during biofilm formation [9]. Other adhesive molecules that ensure the fixation of acinetobacteria in biofilms are the OmpA protein and homologues of staphylococcal Bap proteins (biofilm-associated proteins) [10]. An important element of the structure that unites the biofilm into a single matrix system is the polysaccharide poly-β-(1-6)-N-acetylglucosamine, or PNAG (poly-β-(1-6)-N-acetylglucosamine) [11].

Previously, a system for the express detection of antibiotic resistance for *S. aureus* and *E. coli* using the oscillation mode of an atomic force microscope (AFM) was developed [12,13]. This method is based on the phenomenon of bacterial nanomotion identified by S. Kasas et al. in 2013, which may be determined only by highly sensitive methods. Nanomotion stopped under the influence of antibiotics, to which the bacteria turned out to be sensitive [14]. The great advantage of the new development is its high speed: antibiotic resistance can be detected within an hour after collecting the material, i.e., very quickly compared to the traditional smear for flora and sensitivity to antibiotics, the analysis of which takes at least 48 h. However, it turned out that the developed method is sensitive not only to the detection system, but also to fixing agents that functionalize the cantilever, causing a strong binding of bacteria. The optical method for assessing nanomotion, which S. Kasas created as a development of his brilliant idea, is free from this drawback [15]. The optical method is simpler, more accessible and cheaper. To perform it only a microscope, a camera and a computer program are needed. Thanks to this method, systems have already been developed for assessing the activity of antifungal drugs [16], the viability of only one bacterial cell [17], mitochondrial mobility [18] and the nanomotion of cells of different subtypes of breast cancer [19]. However, in some cases the optical method is quite difficult to apply. In particular, in clinical practice, bacteria often form biofilms. Biofilms have poor optical permeability. In this case, it is easier to determine the nanomotion of bacteria in biofilms by the oscillation method using an AFM.

The purpose of this study was to create a system for the rapid detection of antibiotic resistance in *A. baumannii* 173-p1: (1) to find a fixing agent that ensures reliable fixation of bacteria on the cantilever without loss of their nanomotion; (2) to investigate the nanomotion potential of *A. baumannii* 173-p1 using nutrient-rich and nutrient-limited culture media; (3) to evaluate the possibility of cell nanomotion in biofilms; (4) to determine, using the AFM oscillation method, the antibiotic resistance of the *A. baumannii* 173-p1 strain to antibiotics, in which sensitivity/resistance was previously identified using the classical disk diffusion method.

## 2. Materials and Methods

### 2.1. A. baumannii 173-p1 Culturing

An *A. baumannii* 173-p1 strain was taken from the museum of NSTU named after R.E. Alekseev (this was originally isolated from a lung abscess), and it was routinely grown on Mueller–Hinton agar (FBUN State Scientific Center PNB, Obolensk, Russia) (37 °C, 20 h), then washed off the agar with either normal saline solution (NSS) (FR, MOSFARM LLC, Bogorodskoe, Russia) or lysogeny broth (LB) (Lennox) (Condalab, Madrid, Spain). It was subsequently washed three times with the same medium (450 g, 5 min) and standardized on a spectrophotometer (SPEKS SSP 705, Spectroscopic Systems, Moscow, Russia) (λ = 670 nm) to the suspension optical density of 0.886, which corresponded to a concentration of 10^9^ cells/mL.

### 2.2. Detection of A. baumannii 173-p1 Nanomotion in Different Media

Measuring nanomotion in the AFM oscillation mode is described in detail in [20] and based on the method first described in [14]. Briefly, a cantilever (C-MSCT, f_0_ 4–10 kHz and k 0.010 N/m, Bruker, Billerica, MA, USA) was preliminary functionalized with one of three types of tested fixing agents (0.01% poly-L-lysine (Sigma, St. Louis, MO, USA), 1% solution of rosin (Connectors LLC, St. Petersburg, Russia) in diethyl ether (Panreac, Barcelona, Spain) or fibronectin (IMTEK, Moscow, Russia) at a concentration of 20 μg/mL). After functionalization, but before applying bacteria, a control DFL signal (the difference signal on the photodiode of the AFM microscope) was recorded in the AFM oscillation mode (SMENA, NT-MDT, Zelenograd, Russia) from the cantilever fixed in the AFM holder and placed in the analytical chamber with nutrient medium. Then 20 μL of a bacterial suspension (10^9^ cells/mL) was applied to a cantilever. After incubation (37 °C, 30 min), an experimental signal was obtained from the cantilever with attached bacteria. The Nova software (version Px.3.4.0 rev 17188, oscillation mode) was used to record and process the signal. The nanomotion of bacteria was studied by filling the analytical chamber with one of the nutrient-rich or nutrient-limited media to detect the metabolic activity that determines the nanomotion of bacteria. The idea that the nanomotion pattern could change depending on the studied microorganism and its metabolic activity was summarized in [21]. Since the metabolic activity of bacteria in NSS was minimal, it was used as a nutrient-limited medium to control the lack of metabolic activity. LB with 0.5% NaCl (Lennox), LB broth with 1% NaCl (Miller) as well as meat-peptone broth (MPB) (FBUN State Scientific Center PNB, Obolensk, Russia) were used as nutrient-rich media in which the metabolic activity of bacteria should be observed.

### 2.3. A. baumannii 173-p1 Biofilm Culturing and Measurement of its Nanomotion

A suspension of *A. baumannii* 173-p1 grown and standardized (10^9^ cells/mL) as described in Section 2.1 was added to a sterile 24-well plate (SPL Life Sciences, Pocheon-si, Republic of Korea) in an amount of 400 μL to 600 μL of MPB medium (because it was empirically obtained that this provided the best conditions for growth of *A. baumannii* 173-p1 biofilms), pipetted and then grown (48 h, 37 °C). Then, using a sterile pipette tip, the biofilm was carefully picked up and transferred to the surface of the cantilever, which was previously functionalized with fibronectin as described in Section 2.2. Before transfer of the biofilm, the oscillations of the functionalized cantilever were recorded to obtain control values. After fixation of the biofilm, the cantilever was placed in the analytical chamber and the oscillation signal of the biofilm nanomotion was recorded. To process all control and experimental signals obtained in the oscillation mode, Origin 8 software (OriginLab, Northampton, MA, USA) was used. Frequencies corresponding to external signals were removed from the obtained data array using Fast Fourier Transform (FFT). The lower cutoff limit was set at 2 Hz, since this signal characterizes power grid oscillations and external mechanical oscillations. The upper limit was set at 8 Hz, since “biological” oscillations lie in the range of 1.6–5.1 Hz [22]. The values were normalized by the integral laser signal to exclude differences in the reflectivity of the cantilever surface to achieve a reproducible DFL/Laser ratio. The variance of the cantilever oscillation amplitude (DFL^2^, nm^2^) was used as the comparison index.

### 2.4. Disk Diffusion Test for Study Sensitivity/Resistance of A. baumannii 173-p1 to Antibiotics

A classic disk diffusion test was performed to compare with the results obtained for sensitivity/resistance using the developed express method. 100 μL of bacterial suspension (10^9^ cells/mL) were applied to Petri dishes with sterile Mueller–Hinton agar and seeded with a continuous lawn. Then, a disk with one of the three antibiotics (lincomycin (OAO Dalhimfarm, Khabarovsk, Russia), ceftriaxone (Shreya Life Sciences, Mumbai, India), or doxycycline (OOO Ozon, Zhigulevsk, Russia) at final concentrations of 120 mg/mL, 114 mg/mL and 0.4 mg/mL, respectively) was placed to the surface of the agar in the center of the Petri dish and incubated (20 h, 37 °C). The concentrations of antibiotics were selected in accordance with the recommendations of the company that produces disks for the disk diffusion test (Petritest, Saratov, Russia). The result was obtained on the next day on a matte surface in reflected light. The diameter of growth inhibition zones was measured with an accuracy of 1 mm. The disk diffusion method was performed and analyzed according to the recommendations of EUCAST (version 8.0).

### 2.5. Scanning Electron Microscopy

Scanning electron microscopy (SEM) was used to monitor cell morphology and the affinity of their binding to the cantilever during the selection of the fixing agent as well as to observe the biofilm formation. After the nanomotion-detection experiment, the bacteria were fixed on the cantilever with glutaraldehyde (2.5%, 20 min), the cantilever was washed three times with distilled water, dried in air (24 °C, 60 min) and examined on a JSM-IT300 LV electron microscope (JEOL, Tokyo, Japan) in high vacuum mode at an accelerating voltage of 5 to 20 kV and an electron beam current of no more than 0.2 nA to minimize the effect of the beam on biological samples; the resolution was at least 12 nm. The use of a reduced accelerating voltage of 5 kV allowed for better detailing of the bacterial surface. Electron microscopic images of the cantilever surface were obtained in low-energy secondary electrons [20].

### 2.6. Express Test for Determining the Sensitivity/Resistance of A. baumannii 173-p1 to Antibiotics

The method for determining the sensitivity/resistance of bacteria to antibiotics was first described in [14]. After recording the control (without bacteria) and experimental (with fixed bacteria) values of cantilever oscillations (as it described in Section 2.2), one of the same three antibiotics which were previously used in the disk diffusion test (in final concentrations according to Section 2.4) was added to the analytical chamber, and the DFL signal was recorded. The analytical signal was recorded in 15 min segments during 1 h. A drop in the analytical signal indicated the sensitivity of *A. baumannii* 173-p1 to the antibiotic, while its maintenance at the same level indicated resistance.

### 2.7. Statistical Analysis

Statistical processing was performed using the MatLAB 7.11 platform (MathWorks, Natick, MA, USA). The Lilliefors test showed that the distributions were not normal. Therefore, to compare samples and draw conclusions about the statistical significance of differences, the Kruskal–Wallis test with pairwise comparisons via the Mann–Whitney test was used. The null hypothesis was rejected with a probability of 99.9% (*p* < 0.001).

## 3. Results

### 3.1. Selection of the Optimal Fixing Agent for Fixing Bacteria on a Cantilever

While selecting the fixing agent, the following properties were taken into account: (1) the absence of their own antibiotic activity; (2) the affinity of bacterial fixation to the cantilever; (3) the absence of affection on bacterial nanomotion during the experiment (it was assessed by the stability of the expressed oscillation signal during the experiment with bacteria). From Figure 1 it can be seen that poly-L-lysine is not suitable as a fixing agent, since a small zone of no growth (12 ± 1 mm) was observed (Figure 1A), while rosin (Figure 1B) and fibronectin (Figure 1C) did not demonstrate antibiotic activity against *A. baumannii* 173-p1.

A study of the bacteria morphology on the cantilever surface after the experiment using the SEM method (Figure 2) showed somewhat different results: the morphology changed after using poly-L-lysine only in a small number of cells, while bacteria maintained the ability to divide (Figure 2A). In the case of using rosin and especially fibronectin, significant shielding of bacterial cells by organic fixatives was observed, which makes it much more difficult to clearly visualize the cell morphology (Figure 2B,C). However, the cells also retained their integrity and, judging by their mutual arrangement, their ability to divide. Such coatings obviously have a distinct advantage, since they prevent bacteria desorption from the cantilever surface.

The DFL signal stabilities during the detection of *A. baumannii* 173-p1 nanomotion in the oscillation with different fixing agents are shown in Figure 3.

A stable signal from *A. baumannii* 173-p1 nanomotion was obtained only by fixation with fibronectin, whereas for poly-L-lysine the signal lost stability at the end of the observation, and for rosin at the beginning of the observation.

### 3.2. Evidence That Nanomotion Is Driven by Bacterial Metabolic Activity

The activity of bacterial nanomotion in different nutrient media is shown in Figure 4.

The most preferable medium for bacterial nanomotion was MPB; statistically significant differences were also observed between NSS and both LB broths, with a higher salt concentration LB (Miller) and with a lower salt concentration LB (Lennox).

A similar result was obtained in the SEM study (Figure 5). In the case of bacterial nanomotion detection in MPB, the holistic structure of the cells was observed. Additionally, a constriction was found, indicating that the cell was dividing (Figure 5A). In the case of using LB (Miller), the cells were also actively dividing, but minor indentations in the cell wall were observed (Figure 5B).

### 3.3. Study of Bacterial Nanomotion in Biofilms

Liquid medium-grown *A. baumannii* 173-p1 biofilm that was transferred to the surface of the cantilever with a sterile pipette tip kept its structure as shown using SEM (Figure 6). Excess liquid in the tip of dispenser was carefully removed with filter paper. This methodology was used because attempts to grow biofilm directly on the cantilever surface resulted in its complete overgrowth on both sides of cantilever, which requires separation of the excessive part of biofilm from the cantilever’s reflective side. It either led to the breakdown of the cantilever, or, in the case of successful separation, remains of the biofilm on the upper surface resulted in laser scattering and the presence of a noise signal that could not be interpreted as nanomotion.

### 3.4. Detection of Nanomotion Depended on the Density of the Biofilm

In the case of a very dense biofilm, bacterial nanomotion was not detected (Figure 7). However, if the biofilm was friable, bacterial nanomotion was detected, although during long-term observations, the friable biofilm was gradually “washed off” from the cantilever surface, which was detected by a decrease in the amplitude of cantilever oscillations (Figure 8).

### 3.5. Testing of A. baumannii 173-p1 Sensitivity/Resistance to Antibiotics

To test the sensitivity/resistance of *A. baumannii* 173-p1 to antibiotics, a classic disk diffusion test was firstly used for comparison with the results subsequently obtained by the express method utilizing the oscillation mode of AFM. The results are shown in Figure 9. When using lincomycin, the zone of inhibition of bacterial growth was absent; when using a disk with ceftriaxone, the zone of growth inhibition was 12 ± 1 mm, and when using doxycycline, the zone of growth inhibition was 20 ± 1 mm.

Similar results were obtained by studying the morphology of bacteria using the SEM method after incubation in an analytical chamber with an antibiotic (Figure 10). After incubation with lincomycin, the morphology of bacterial cells was practically unchanged; in addition, they retain the potential for active division (Figure 10A); after incubation with ceftriaxone, under the influence of which an insignificant zone of growth inhibition was detected for *A. baumannii* 173-p1, the potential for division was preserved, but the integrity of the bacterial cell wall was disrupted, and the bacteria themselves were deformed (Figure 10B). After incubation with doxycycline, to which, according to the disk diffusion method, the bacteria were found to be sensitive, the bacteria died in large quantities, with the release of intracellular contents (Figure 10C).

Absolutely identical results were obtained by the developed method for express diagnostics of *A. baumannii* 173-p1 antibiotic resistance (Figure 11). In the case of bacteria incubation with antibiotics to which *A. baumannii* 173-p1 were resistant, i.e., with lincomycin and ceftriaxone, the variances of the oscillation amplitudes of bacteria and bacteria incubated with the antibiotic statistically significantly increased (*p* < 0.001) (Figure 11A,B,D). In the case of incubation of *A. baumannii* 173-p1 with doxycycline, to which the bacterium was sensitive according to the results of the disk diffusion test, a statistically significant decrease in the variance of the cantilever oscillation amplitude was observed (*p* < 0.001) (Figure 11C).

A comparison of the classical method for detecting of bacterial antibiotic resistance using the disk diffusion method with a new express method based on the detection of bacterial nanomotion is summarized in Table 1.

## 4. Discussion

The detection of own-antibiotic activity of poly-*L*-lysine against *A. baumannii* 173-p1 makes this fixation agent unsuitable for determining the sensitivity/resistance of bacteria to antibiotics, since it could give a false-positive result. Although SEM studies did not show any significant change in the morphology of bacterial cells, and even demonstrated the preserved ability of cell division, the results obtained from measuring the DFL signal on AFM suggested that poly-*L*-lysine is indeed unsuitable for fixation, since the signal decreases slightly by the end of the observation. This can be explained both by the presence of weak antibiotic activity of poly-*L*-lysine and by the washout of the fixation agent by the nutrient medium during cantilever oscillation, which leads to desorption of bacteria from the surface. The second option is more probable, since, on the contrary, when fixing with rosin, a weak signal was initially observed, probably due to the fact that rosin is a too strong a fixative, capable of limiting bacterial nanomotion, but the dilution effect leads to the removal of this limitation and a significant increase in the signal. However, such signal suppression at the initial stages of the study can lead to false-negative results, therefore rosin is also not suitable for fixing *A. baumannii* 173-p1. Only fibronectin gave a stable, reproducible result; therefore, fibronectin should be used as a fixing agent for detecting the sensitivity/resistance of *A. baumannii* 173-p1.

The absence of bacterial nanomotion in a nutrient-limited NSS (0.9% NaCl), which did not provide *A. baumannii* 173-p1 with nutrients, proves that bacterial nanomotion correlates with its metabolic activity. Bacterial metabolism, actively triggered by LB, and especially by MPB, is characterized by the appearance of active bacterial nanomotion, which was recorded in the oscillation mode (by the DFL signal). Similar conclusions were reached by Willaert et al. (2020), who inhibited the motility of yeast cells with a nutrient-limited phosphate buffer saline and completely blocked it with ethanol [15].

The greatest number of difficulties were observed in the study of nanomotion of bacteria in biofilms. In particular, the study revealed that the amplitude of *A. baumannii* 173-p1 nanomotion is determined by both the nature of biofilms (in particular, their density) and the method of sample preparation. But, in any case, the amplitude of bacterial nanomotion in biofilms is much smaller than in suspension forms. This may be due to a number of reasons: (1) the polysaccharide–protein matrix can “constrain” the movements of bacteria; (2) the biofilm matrix can increase the surface tension on one side of the cantilever and, thus, reduce the sensitivity of the transducer; (3) one can also assume a general decrease in the metabolic activity of each individual bacterium due to the presence of “quorum sensing”. McQueary and Actis’s work [23] showed that twitching (the main mechanism of *Acinetobacter* movement) does not affect the intensity of biofilm formation; however, in all likelihood, the formation of nanofilms directly affects the intensity of bacterial nanomotion.

Oscillations due to bacterial nanomotion are characterized by a statistically significant increase in amplitude in the case of bacterial resistance to antibiotics. A similar observation was made by Venturelli et al. (2020) [24], who also studied bacterial resistance to antibiotics using the AFM oscillation mode. Since bacterial nanomotions are primarily due to their metabolic activity, we associate the increase in oscillation amplitude with the activation of mechanisms that cause resistance; in particular, with the activation of pumps, modification of the drug target, drug inactivation, target switching or its sequestration, which increase oscillations from the baseline level of metabolic activity. In contrast, in the case of *A. baumannii* 173-p1 treatment with doxycycline, a statistically significant drop in oscillation amplitude was observed, indicating the cessation of metabolic activity of bacteria and their death.

It is important that all three methods—the classical disk diffusion method, the method of studying the morphology of bacteria using SEM and the AFM-based express method of detection—showed absolute convergence of the results obtained. However, the use of express testing of antibiotic resistance using the AFM oscillation mode has two most significant advantages for clinical practice: high sensitivity and high speed of determining the sensitivity/resistance of bacteria to antibiotics (less than 1 h), unlike other methods (Table 1). This gives the possibility to use this method to switch from empirical to etiotropic therapy in a short period of time.

## 5. Conclusions

It has been shown that bacterial nanomotion, which can be detected using the AFM oscillation mode both for bacterial suspension form and biofilm, is caused by the metabolic activity of bacteria. At the same time, an appropriate fixing agent (in this case fibronectin) is crucial for obtaining a stable and pronounced analytical signal, which is important for further explorations. The sensitivity/resistance to antibiotics results from the disk diffusion test, SEM morphology study and AFM oscillation study are consistent with each other. The detection of nanomotion has great clinical significance, since it allows one to draw a conclusion about the sensitivity/resistance of bacteria to antibiotics in a short time (within an hour) and will allow one to quickly determine antibiotic therapy in critical cases for such a clinically significant microorganism as *Acinetobacter baumannii*.

## Figures and Tables

**Figure 1 biomedicines-12-02034-f001:**
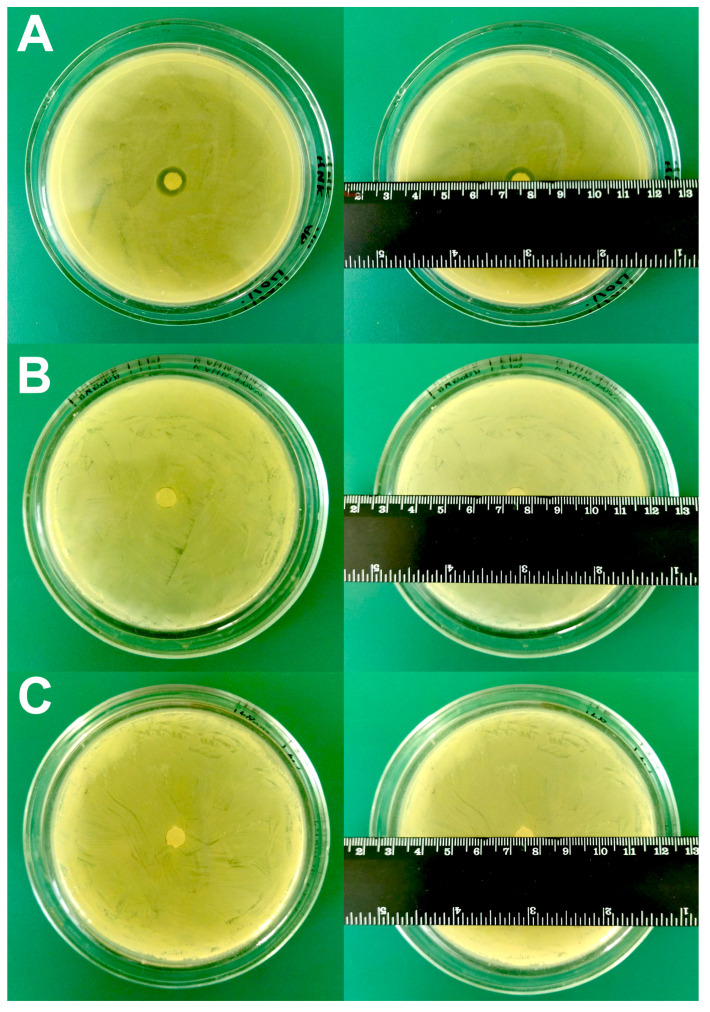
Study of growth inhibition zones for different types of fixing agents. (**A**) 0.01% poly-L-lysine. (**B**) 1% rosin solution in diethyl ether. (**C**) 20 μg/mL fibronectin.

**Figure 2 biomedicines-12-02034-f002:**
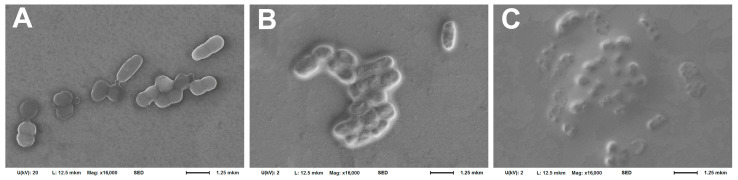
Morphology of *A. baumannii* 173-p1 fixed on the cantilever surface with 2.5% glutaraldehyde and examined by SEM after using different cantilever fixatives. (**A**) 0.01% poly-L-lysine. (**B**) 1% rosin solution in diethyl ether. (**C**) 20 μg/mL fibronectin.

**Figure 3 biomedicines-12-02034-f003:**
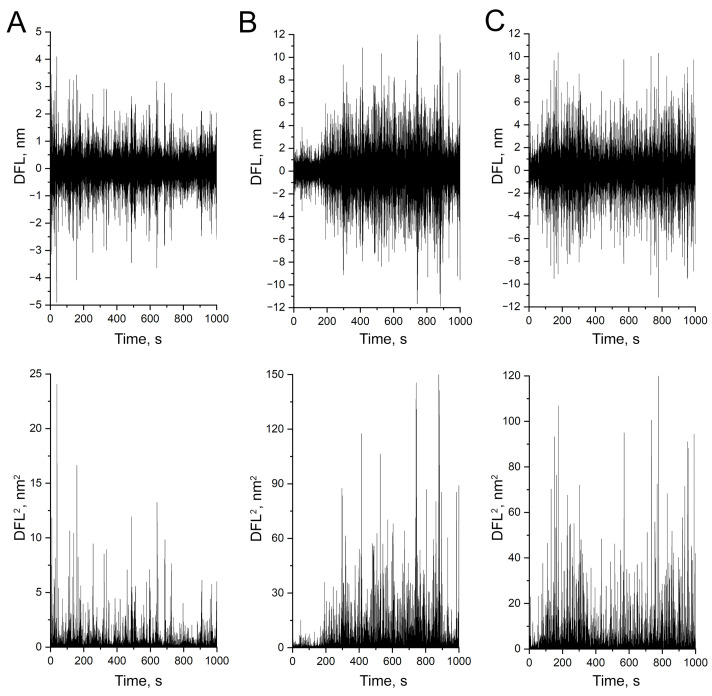
DFL signal characterizing bacterial nanomotion on different fixatives: up: after Fast Fourier transform, down: variances of oscillation amplitudes. (**A**) 0.01% poly-L-lysine. (**B**) 1% rosin solution in diethyl ether. (**C**) 20 μg/mL fibronectin.

**Figure 4 biomedicines-12-02034-f004:**
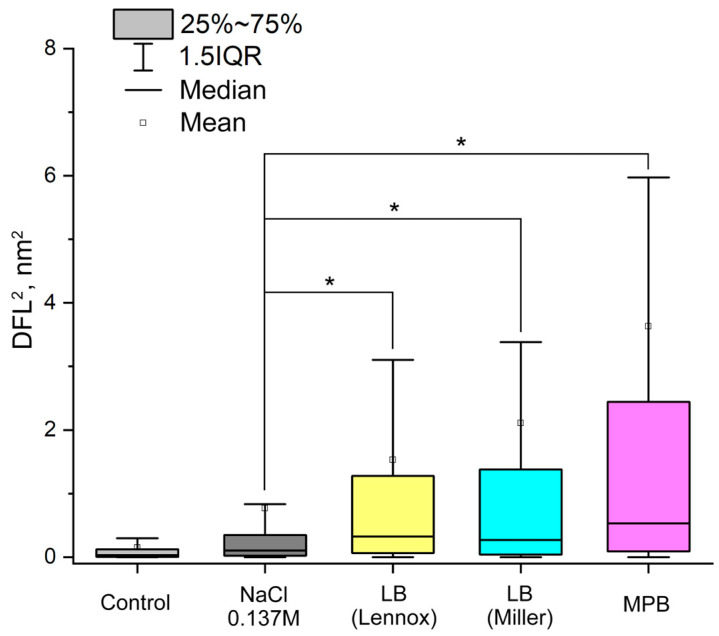
Changes in the nature of *A. baumannii* 173-p1 nanomotion, depending on cultivation on a nutrient-limited 0.9% NaCl (normal saline solution and nutrient-rich media): LB broth (Lennox) with 0.5% NaCl, LB broth (Miller) with 1% NaCl and MPB (meat-peptone broth (*—differences are statistically significant, *p* < 0.001)). Variance (DFL^2^, nm^2^) was used as a statistically processed signal, the results demonstrate a series of five experiments for each medium.

**Figure 5 biomedicines-12-02034-f005:**
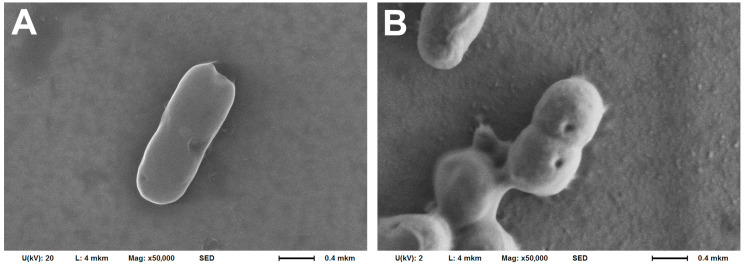
SEM images of *A. baumannii* 173-p1 after nanomotion-detection experiments in different media. (**A**) MPB. (**B**) LB (Miller).

**Figure 6 biomedicines-12-02034-f006:**
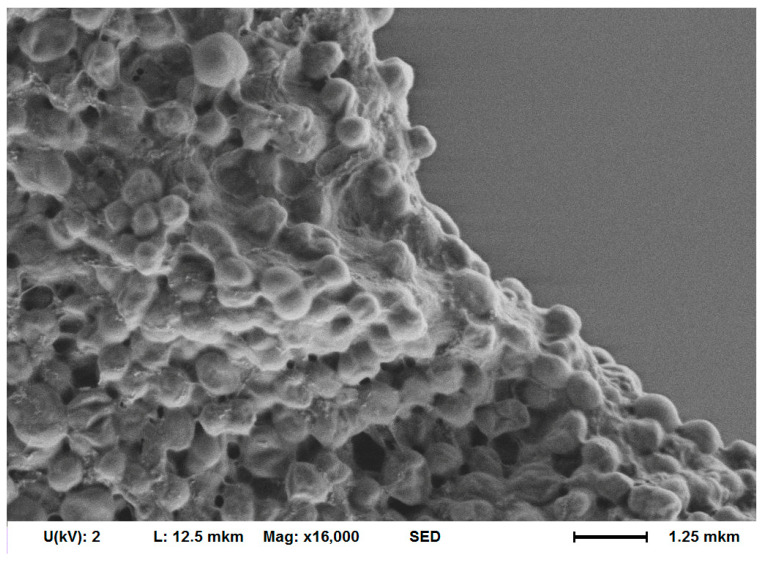
SEM images of *A. baumannii* 173-p1 biofilm on the cantilever surface (fixed by 2.5% glutaraldehyde).

**Figure 7 biomedicines-12-02034-f007:**
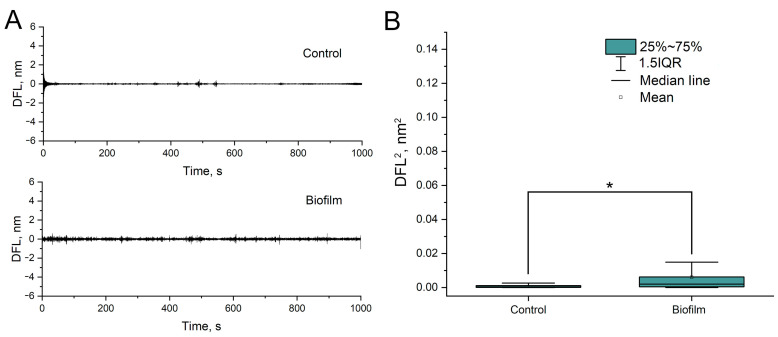
Nanomotion of *A. baumannii* 173-p1 inside of density biofilm. (**A**) DFL signal after Fast Fourier transform (up: control-cantilever without biofilm, down: cantilever with biofilm). (**B**) Statistical analysis of oscillation amplitudes variances (*—differences are statistically significant, *p* < 0.001).

**Figure 8 biomedicines-12-02034-f008:**
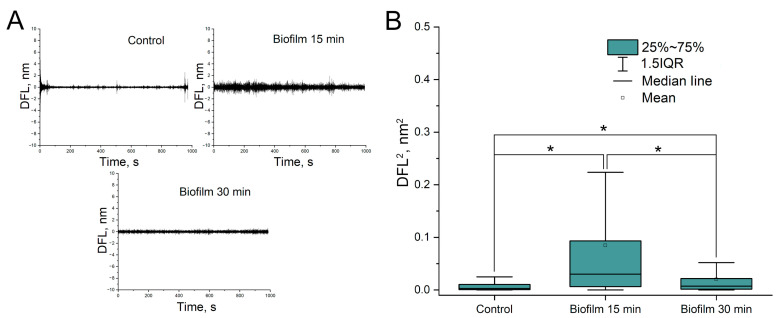
Nanomotion of *A. baumannii* 173-p1 inside of friable biofilm. (**A**) DFL signal after Fast Fourier transform (up: control-cantilever without biofilm, down: cantilever with biofilm). (**B**) Statistical analysis of oscillation amplitudes variances (*—differences are statistically significant, *p* < 0.001).

**Figure 9 biomedicines-12-02034-f009:**
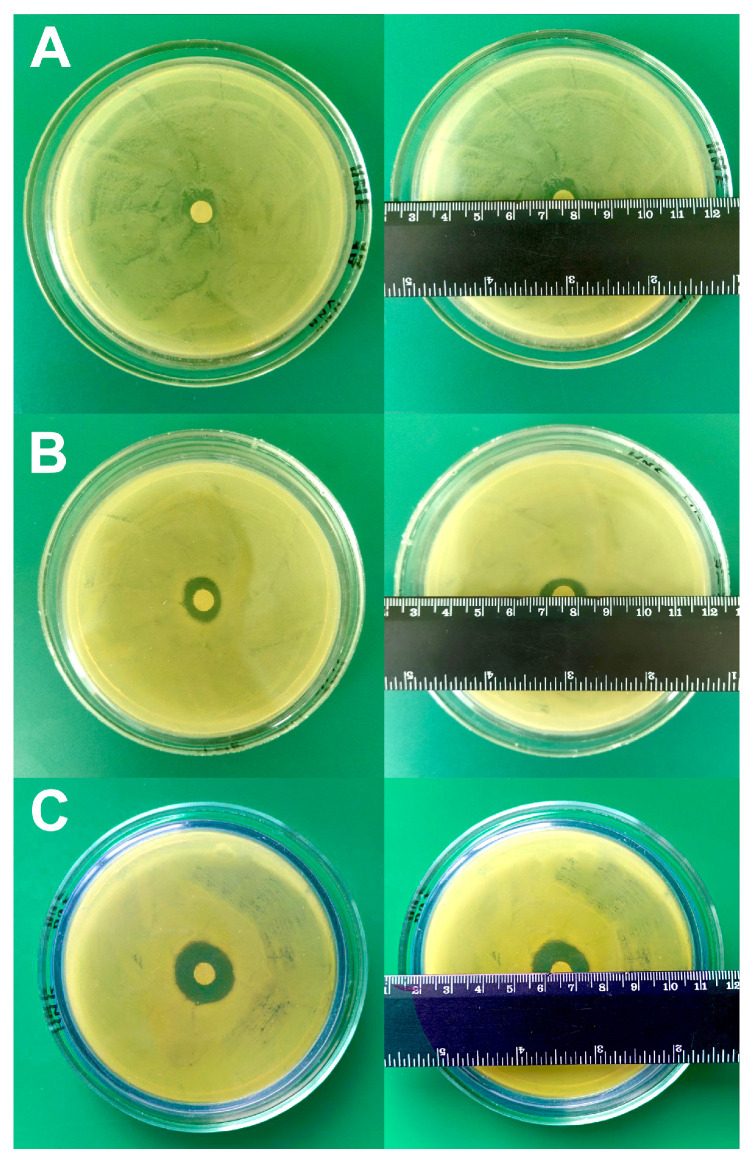
Determination of *A. baumannii* 173-p1 growth inhibition zones under the influence of different antibiotics using the disk diffusion method. (**A**) Lincomycin. (**B**) Ceftriaxone. (**C**) Doxycycline.

**Figure 10 biomedicines-12-02034-f010:**
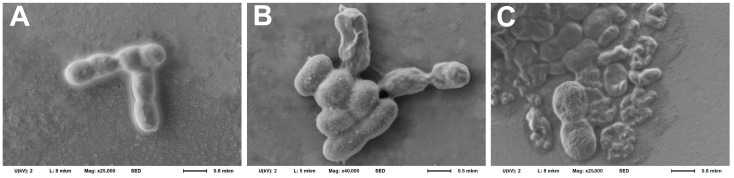
SEM images of *A. baumannii* 173-p1 fixed on a C-MSCT cantilever with 2.5% glutaraldehyde after DFL signal recording under influence of different antibiotics. (**A**) Lincomycin. (**B**) Ceftriaxone. (**C**) Doxycycline.

**Figure 11 biomedicines-12-02034-f011:**
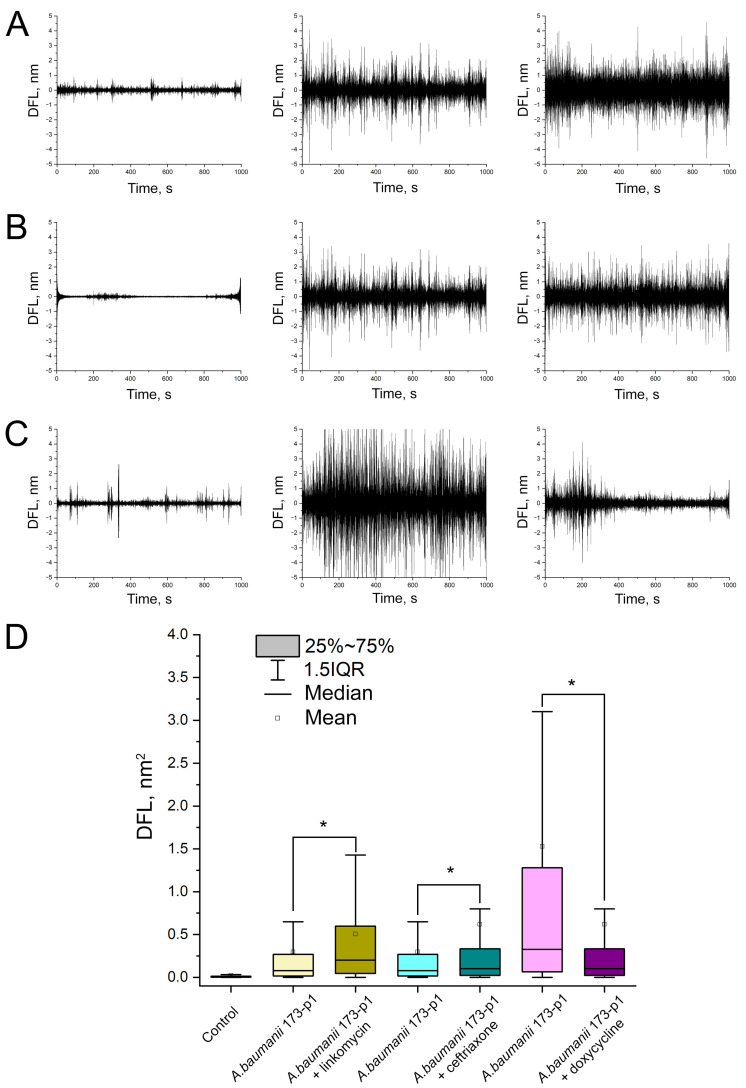
Detection of *A. baumannii* 173-p1 antibiotic resistance based on bacterial nanomotion, utilizing the DFL signal after Fast Fourier transform. (**A**) Changes in the amplitude of oscillations in response to lincomycin: the first column shows control oscillations of the functionalized cantilever, the second column shows control oscillations of the functionalized cantilever with bacteria, the third column shows experimental oscillations after immersion of the cantilever in a chamber with an antibiotic. (**B**) Changes in the amplitude of oscillations to ceftriaxone. (**C**) Changes in the amplitude of oscillations to doxycycline. (**D**) Results of statistical processing of obtained oscillation amplitude variances for all studied antibiotics in comparison (*—differences are statistically significant, *p* < 0.001).

**Table 1 biomedicines-12-02034-t001:** The difference between disk diffusion and AFM-oscillation method detection of sensitivity/resistance bacteria to antibiotics.

Criterion	Disk Diffusion Method	Oscillation Method
Availability of an official standard	EUCAST	-
Current price	Lower	Higher
Reproducibility	+	+
Sensitivity	Low	High
Time of analysis	48 h	Less than 1 h

## Data Availability

All data are available through the correspondence author.

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
