# Peer review of "Rapid Detection of Acinetobacter baumannii Suspension and Biofilm Nanomotion and Antibiotic Resistance Estimation"

_biomedicines, 2024, doi:10.3390/biomedicines12092034_

Round 1

Reviewer 1 Report

Comments and Suggestions for Authors

1. There is little citations in the methodology section, but the reviewer does not think all the methods are completely novel. Please provide citations based on which the methods were designed or modified.

2. Line 220 Please try do not use "definite" statements like "it is obvious" as they should be avoided in scientific language. Use alternative statements that describe your observations.

3. Line 245 Bacterial nanomontion were not detected - Bacterial nanomotion WAS not detected or bacterial nanomotionS were not detected

4. In regard to antibiotic resistantce experiment: why lincomycin, ceftriaxone and doxycycline were chosen? Do the authors have the results of the antibiotic resistance tests for the strain? If so providing those data in the methodology section would help the readers to understand the strain antibiotic sensitiviti/resistance and facilitate the understnading why such antibiotics were chosen for the test.

Comments on the Quality of English Language

The English language of this manuscript is fine overall. Minor editing regarding is/are, was/were etc. is required.

Author Response

Dear colleague, we express our sincere gratitude to you for your work with our manuscript. It is thanks to your highly competent comments we have done an additional job and we hope that now the manuscript has become much better. We also want to thank you for your kind attitude.

  1. There is little citations in the methodology section, but the reviewer does not think all the methods are completely novel. Please provide citations based on which the methods were designed or modified.

Thank you for your comment. We fully agree that in Materials and Methods we did not sufficiently cite the articles of the researchers who developed the methods. We have inserted additional references. We did not insert references in two paragraphs: (1) 2.4. Disk-diffusion test for study sensitivity/resistance of A. baumannii 173-p1 to antibiotics because in this paragraph we gave link to International Standard – EUCAST and (2) 2.3. A. baumannii 173-p1 biofilm culturing and measurement of its nanomotion because it was a method that we had been developing ourselves for just long time.

  1. Line 220 Please try do not use "definite" statements like "it is obvious" as they should be avoided in scientific language. Use alternative statements that describe your observations.

Thank you. We change style and we will try to avoid such words in the future.

  1. Line 245 Bacterial nanomotion were not detected - Bacterial nanomotion WAS not detected or bacterial nanomotionS were not detected.

Sorry for this gross mistake in English - we have corrected it.

  1. In regard to antibiotic resistance experiment: why lincomycin, ceftriaxone and doxycycline were chosen? Do the authors have the results of the antibiotic resistance tests for the strain? If so providing those data in the methodology section would help the readers to understand the strain antibiotic sensitivity/resistance and facilitate the understanding why such antibiotics were chosen for the test.

Thank you for your comment. In fact, Acinetobacter is one of the most difficult microorganisms to work with, since it is a type of superbacterium which is resistant to the vast majority of antibiotics. Therefore, we first did many tests using the classic disk diffusion method (the results are shown in Fig. 9). Then we selected three antibiotics based on the principle of using: one antibiotic to which the bacteria are completely resistant, a second antibiotic to which the bacteria will be sensitive, and a third antibiotic that occupies an intermediate position. It was interesting to track whether the oscillation method would be sensitive enough to detect these differences. Fortunately, these differences were seen in both methods. Actually, we also did a kind of check - we asked the bacteriology laboratory at the hospital to do a disk diffusion test with these antibiotics so that we could be sure of the correctness of our results. This kind of external control showed similar results, so we have since then calmly used these antibiotics for all our experiments.

The English language of this manuscript is fine overall. Minor editing regarding is/are, was/were etc. is required.

Thank you for your comment. We have edited the manuscript and hope that the manuscript is now easier to read. We have also hugely rearranged materials and method section to make it structure more logical.

Reviewer 2 Report

Comments and Suggestions for Authors

This study developed a rapid method for detecting antibiotic resistance in Acinetobacter baumannii using atomic force microscopy (AFM). The method leverages bacterial nanomotion, which is linked to metabolic activity, to assess antibiotic sensitivity. The results demonstrate that AFM provides comparable results to traditional methods but with significantly faster turnaround times, enabling rapid diagnosis of antibiotic resistance. The manuscript is written well and effectively conveys its primary findings. However, there are some recommendations that increase the quality of the manuscript.

1.     The title of the manuscript may be written more concise manner like “Rapid detection of Acinetobacter baumannii suspension and biofilm nanomotion, and antibiotic resistance estimation”

2.     Authors may replace the terms “Acinetobacter baumannii, nanomotion, antibiotic resistance” from the keywords section as these terms are already mentioned in the title of the manuscript.

3.     The authors notes difficulties in preparing samples for biofilm testing on the cantilever, such as complete overgrowth on both sides of the cantilever or laser scattering due to remaining biofilm after separation. These challenges could affect the accuracy and reliability of the results​.

4.    While the oscillation method using Atomic Force Microscopy (AFM) is effective for detecting bacterial resistance, it is less practical than the optical method in some clinical settings, especially for biofilms that are optically impermeable.

Author Response

Dear colleague, we express our sincere gratitude to you for your work with our manuscript. It is thanks to your highly competent comments we have done an additional job and we hope that now the manuscript has become much better. We also want to thank you for your kind attitude.

This study developed a rapid method for detecting antibiotic resistance in Acinetobacter baumannii using atomic force microscopy (AFM). The method leverages bacterial nanomotion, which is linked to metabolic activity, to assess antibiotic sensitivity. The results demonstrate that AFM provides comparable results to traditional methods but with significantly faster turnaround times, enabling rapid diagnosis of antibiotic resistance. The manuscript is written well and effectively conveys its primary findings. However, there are some recommendations that increase the quality of the manuscript.

  1. The title of the manuscript may be written more concise manner like “Rapid detection of Acinetobacter baumannii suspension and biofilm nanomotion, and antibiotic resistance estimation”.

Thank you very much. Your title is much better and shorter. We have changed the title.

  1. Authors may replace the terms “Acinetobacter baumannii, nanomotion, antibiotic resistance” from the keywords section as these terms are already mentioned in the title of the manuscript.

Thank you for your great tips, thanks to you we have expanded the range of keywords. And we will use this method in the future.

  1. The authors notes difficulties in preparing samples for biofilm testing on the cantilever, such as complete overgrowth on both sides of the cantilever or laser scattering due to remaining biofilm after separation. These challenges could affect the accuracy and reliability of the results​.

Thank you for your comment. Actually, we have made these comments to inform other researchers about possible difficulties in the case of working with biofilms. If someone works with them, so that they do not repeat these approaches. Initially, we tried to grow films directly on the cantilever and the biofilms grew on the entire cantilever. Because of this, we doubted the correctness of the results due to the phenomenon of laser scattering. Then we developed two different methods and compared them: (1) an option when we cut a suppository and pressed the cantilever into it. In this case, the biofilm was formed only on one surface – the outer one from the suppository, but sometimes the cantilever broke when released from the suppository (so we don’t think this method is suitable); (2) an option when the biofilm was taken with a cut pipette tip and sterilely transferred to the cantilever, which was previously functionalized. Both of these methods with the formation of a biofilm only on one side of the cantilever showed similar results. Since the reflective side of the cantilever remained clear, there was no scattering of the laser, and we obtained reproducible results. Therefore, we are confident in the results we present.

  1. While the oscillation method using Atomic Force Microscopy (AFM) is effective for detecting bacterial resistance, it is less practical than the optical method in some clinical settings, especially for biofilms that are optically impermeable.

We absolutely agree with you. The optical method is much more practical and easier both in sample preparation and in clinical practice. And most importantly, it is much cheaper. We wrote a little about this. We really enjoyed working with the optical method. And we even wanted to show the advantages of the optical method in studying biofilms. But, unfortunately, our attempts to analyze the movements of bacteria in a biofilm using the optical method were smashed by the optical impermeability of the films. We are now trying to make the films "loose", but so far they remain optically impermeable. It can be assumed (but this is only an assumption, not a fact) that capturing nanomotion in biofilms will also not be easy, even if we achieve such a biofilm consistency that will give us this opportunity. It is simple and the oscillation method shows that the nanomotion of bacteria in biofilms is much less pronounced than in the suspension form. In addition, some dependence of bacterial nanomotion on the density of the biofilm has been revealed.
